# Targeted Metabolomics Study on the Effect of Vinegar Processing on the Chemical Changes and Antioxidant Activity of *Angelica sinensis*

**DOI:** 10.3390/antiox12122053

**Published:** 2023-11-28

**Authors:** Linlin Chen, Long Li, Fengzhong Wang, Shenghai Hu, Tingting Ding, Yongru Wang, Yulong Huang, Bei Fan, Jing Sun

**Affiliations:** 1Key Laboratory of Agricultural Products Quality and Safety Collection, Storage and Transportation Control, Ministry of Agriculture and Rural Affairs, Institute of Food Science and Technology, Chinese Academy of Agricultural Sciences, Beijing 100193, China; chenlinlin@luibe.edu.cn (L.C.); lilong02@caas.cn (L.L.); wangfengzhong@caas.cn (F.W.); 82101232168@caas.cn (T.D.); 202354103@bucm.edu.cn (Y.W.); 2School of Pharmacy, Harbin University of Commerce, Harbin 150010, China; 3Agricultural Product Storage and Processing Research Institute, Gansu Academy of Agricultural Sciences, Lanzhou 730070, China; hushenghai1972@126.com (S.H.); huangyulong@gsagr.cn (Y.H.)

**Keywords:** *Angelica sinensis*, vinegar processing, chemical composition, antioxidant activity, metabolomics

## Abstract

*Angelica sinensis* (Oliv.) Diels (*A. sinensis*) has a long processing history. In order to obtain a more valuable composition and higher antioxidant behavior, it is often processed by stir-frying and vinegar treatment. However, the underlying mechanism of chemical changes remains ambiguous. Using UPLC-QQQ-MS/MS alongside targeted metabolomics techniques, this study probed the variances between crude and processed *A. sinensis*. We identified 1046 chemical components in total, 123 differential components in stir-fried *A. sinensis,* and 167 in vinegar-treated ones were screened through multivariate statistical analysis. Moreover, 83 significant compounds, encompassing amino acids, phenolic acids, etc., were identified across both processing methods. The in vitro antioxidant activities of these *A. sinensis* forms were assessed, revealing a positive correlation between most of the unique components emerging after processing and the antioxidant capabilities. Notably, post-processing, the chemical composition undergoes significant alterations, enhancing the antioxidant activity. Specific compounds, including 4-hydroxybenzaldehyde, syringetin-3-*O*-glucoside, and salicylic acid, greatly influence antioxidant activity during processing.

## 1. Introduction

China stands as a pioneer in spice utilization, producing nearly 200 spice varieties [1,2]. These spices are essential in various culinary applications, from home cooking to food processing. Technological advancements have introduced diverse processing techniques like dehydration and both high and low-temperature processing [2,3]. Processing can influence the moisture content, sensory attributes, and biological activity of spices [4], often enhancing taste, texture, and nutrition [5]. The exact impact of these methods on the chemical composition is still under exploration, spurring investigations into the mechanisms of these material changes.

*Angelica sinensis* (Oliv.) Diels (*A. sinensis*), a perennial herbaceous plant from the Umbelliferae family (named Danggui in China), holds a significant place in China’s culinary history. It is both a revered herbaceous plant and a culinary ingredient. Historically, its functional applications were used to improve blood circulation and treat menstrual irregularities [6]. It was treated as a spice, essential oil, and therapeutic agent in Europe and the U.S., further underscoring its significance [7]. The literature abounds with high-temperature processing methods for *A. sinensis*, evolving from simple techniques to more complex processes like stir-frying with auxiliary materials [8]. Despite the extensive documentation on the processing with wine, soil, and charcoal, limited insights exist on the fried and vinegar variants, especially regarding the alterations in chemical constituents. 

Metabolomics studies, which focus on an organism’s metabolites, were used to offer a perspective on understanding the relationships and functions within vast biological datasets [9,10]. This approach can be used to facilitate a deeper comprehension of metabolic processes, response mechanisms, and safety evaluations [11]. Given the potential of metabolomics in nutritional and pharmaceutical research [12], it can provide a comprehensive chemical and biological profile of the subject under study. 

Some insights into the effects of processing on A. sinensis’s chemical composition were offered by the existing literature, but a deep dive into the chemical complexity of post-processing remains uncharted, particularly for stir-fried and vinegar-processed variants. Given this backdrop, our study delves into the changes in the chemical compositions of these two variants. Utilizing ultra-high-performance liquid chromatography triple quadrupole mass spectrometry (UPLC-MS/MS) and targeted metabolomics, we aim to understand the processing mechanisms of *A. sinensis*, offering a foundation for its broader culinary and medicinal applications.

## 2. Materials and Methods

### 2.1. Chemicals and Reagents

Methanol (MeOH, LC-MS grade) and acetonitrile (ACN, LC-MS grade) were both purchased from Merck Co., Ltd. (Shanghai, China). Formic acid (AC, LC-MS grade) was purchased from Aladdin Biochemical Technology Co., Ltd. (Shanghai, China). Purified water was obtained from the Milli-Q^®^ HX SD water purification system, which was provided by Merck Co., Ltd. (Shanghai, China). DPPH free radical-scavenging assay kit, hydroxyl free radical-scavenging assay kit, and total antioxidant capacity (T-AOC) assay kit were purchased from Shanghai Yuanye Biotechnology Co., Ltd. (Shanghai, China).

A high-speed Chinese medicine microtome (H1043, Shenzhen DEKE Machinery Technology Co. Ltd., Shenzhen, China), freeze dryer (Scientz-100F, Henan BROTHER INSTRUMENT Equipment Co. Ltd., Zhengzhou, China), Electronic balance (MS105DU, Shenzhen SANLI Technology Co., Ltd., Shenzhen, China), mixing grinder (MM400, Germany Retsch, Haan, Germany), high-speed centrifuge (75007201, Thermo Fisher Scientific, Rodano, Italy), multifunctional microplate reader (SpectraMax M, Meigu Molecular Instruments (Shanghai) Co., Ltd., Shanghai, China) were used. Ultra-high-performance liquid chromatography (UPLC, ExionLCTM AD, AB sciex, Framingham, MA, USA. https://sciex.com.cn/, accessed on 12 December 2022) and tandem triple quadrupole mass spectrometry (MS/MS, Applied Biosystems 4500 QTRAP, AB sciex, Framingham, MA, USA. https://sciex.com.cn/, accessed on 12 December 2022) were used. 

### 2.2. Collection and Processing of A. sinensis Samples and Preparation of Test Solution

#### 2.2.1. Fresh *A. sinensis* Material Samples Collection and Processing

In October 2022, sufficient biological replicate samples of fresh *A. sinensis* were collected from Tianquanshe Village, Dingjiashan Village, New Town, Lintan County, Linxia Prefecture, Gansu Province, China. The collected *A. sinensis* samples were identified by Professor Pengfei Tu from Peking University. The fresh *A. sinensis* medicinal materials were removed, washed, and cut into thin pieces, and after vacuum freeze-drying, the raw *A. sinensis* (A) pieces were prepared (Figure 1 and Appendix A) [13]. 

Stir-fried *A. sinensis* (S-A): A certain amount of *A. sinensis* tablets were taken out, which were fried at 120 °C for 6.5 min until the slices turned yellow, and then the samples were taken out of the pan. After spreading out and cooling, the stir-fried *A. sinensis* pieces were finished (Figure 1 and Appendix A). 

Vinegar-made *A. sinensis* (V-A): An appropriate amount of *A. sinensis* tablets was added by 15% rice vinegar and moistened at room temperature for 12 h. The tablets were stir-fried at 120 °C for 6.5 min until the color was deepened. After slightly burned spots appeared on the slices, we let them cool down to obtain vinegar *A. sinensis* tablets (Figure 1 and Appendix A) [14]. Subsequently, all the treated *A. sinensis* samples were ground into a homogeneous powder. 

#### 2.2.2. Preparation and Extraction of Processed Samples of *A. sinensis*

Three batches of A, S-A, and V-A sample powders were selected. A total of 0.5 g of each was accurately weighed by electronic balance, and 12 mL of 70% methanol solution was added and soaked for 4 h. The sample solution was extracted via ultrasound twice for 60 min each time; samples were centrifuged at 4 °C and 5180× *g* for 6 min. Then, the supernatant was filtered with a microporous filter membrane (0.22 μm Pore Size) to obtain A, S-A, and V-A test solutions (Appendix A). The test solutions were saved at 4 °C for subsequent experimental analysis. In the pre-experiment phase, the effects of different extraction solvents, including 90%, 70%, 50%, and 30% methanol, on the extraction yield of the chemical components of *A. sinensis* were investigated. The results showed that 70% methanol could better dissolve the chemical components of *A. sinensis* and had a higher content. Therefore, we used 70% methanol as the solvent.

#### 2.2.3. Preparation of Quality Control (QC) Sample and System Stability Investigation

QC sample was prepared and mixed well by accurately pipetting 40 μL of each test solution [15]. QC sample was used to monitor the stability and repeatability of instruments and systems. Before the chromatographic and mass spectrometry analysis of *A. sinensis* samples, the same QC sample was run 6 times in a row to balance the system and maintain its stability. In the detection process of *A. sinensis* and its processed products, QC samples were run once for every 6 samples. The stability of the system and samples was examined by observing the LC-MS ion chromatogram combined with multivariate statistical analysis.

### 2.3. Chromatography and Mass Spectrometric Analysis

The sample extracts were analyzed using a UPLC-ESI-MS/MS system (UPLC, ExionLCTM AD, AB sciex, Framingham, MA, USA. https://sciex.com.cn/, accessed on 12 December 2022; MS, Applied Biosystems 4500 Q TRAP, AB sciex, Framingham, MA, USA. https://sciex.com.cn/, accessed on 12 December 2022). The analytical conditions were as follows.

#### 2.3.1. UPLC Conditions

The chromatographic separations were conducted on an Agilent SB-C18 (1.8 μm, 2.1 mm × 100 mm, Agilent Technologies, Santa Clara, CA, USA) column. Column oven was maintained at 40 °C, and the elution was achieved using 0.1% formic acid (A) and acetonitrile with 0.1% formic acid (B). Sample measurements were performed with a gradient program that employed the starting conditions of 95%A, 5% B. Within 9 min, a linear gradient to 5% A, 95% B was programmed, and a composition of 5% A, 95% B was kept for 1 min. Subsequently, a composition of 95% A and 5.0% B was adjusted within 1.1 min and kept for 2.9 min. The flow velocity was set as 0.35 mL per minute. The injection volume was 4 uL. The effluent was alternatively monitored. At the end of each run, the initial composition of the mobile phase (95% A) was permitted to re-equilibrate the whole system for 5 min.

#### 2.3.2. ESI-QTRAP-MS Conditions

The ESI source operation parameters were as follows: source temperature 550 °C, ion spray voltage (S) 5500 V (positive ion mode)/−4500 V (negative ion mode), ion source gas I (GSl), gas II (GSIl), curtain gas (CUR) were set at 50, 60, and 25 psi, respectively, and the collision-activated dissociation (CAD) was high [16]. QQQ scans were acquired as MRM experiments with collision gas (nitrogen) set to medium. DP (declustering potential) and CE (collision energy) for individual MRM transitions were performed with further DP and CE optimization. A specific set of MRM transitions was monitored for each period according to the metabolites included within this period.

### 2.4. Data Acquisition and Processing

The screening database of known ingredients of *A. sinensis* was established based on Metware’s self-built database and the chemical composition of *A. sinensis* in the related literature. According to the data information of the secondary spectrum, the compounds were qualitatively analyzed, and the composition and planar structure of the compounds in *A. sinensis* were determined by obtaining the information of *A. sinensis* compound, such as the fragment ion information of molecular formula, molecular weight, and mass spectrometry. In the qualitative analysis of data, we need to exclude some interference or repetitive signals, such as ammonium ions, potassium ions, sodium ions, fragment ion signals of some large molecular weight substances, as well as isotope-related signals. Mass spectrometry data can be processed using Analyst 1.6.3 software. Then, the mass-to-charge ratio information of *A. sinensis* compounds in the total ion flow map was extracted and combined with the actual MS/MS information, matched and confirmed through a self-built database, and finally, the compound planar structure in *A. sinensis* was characterized. The characteristic ions of *A. sinensis* compounds were then screened and analyzed by MRM mode. Save data on the signal intensity of the characteristic ions of angelica compounds and download the mass spectrometry file. After analyzing the compound mass spectrometry data of different *A. sinensis* samples, the area of the chromatographic peak of *A. sinensis* compound was calculated, and the integration and correction were carried out to express the relative content of the corresponding substance with the peak area of the chromatographic peak. The chemical composition of *A. sinensis* was relatively quantified.

### 2.5. In Vitro Antioxidant Activity Assay

Determination of the DPPH radical scavenging: the experimental method was operated in strict accordance with the corresponding detection kit instructions. A total of 50 μL of nitrogen radical extract and 450 μL of DPPH solution were added to the blank tubes. A total of 50 μL of *A. sinensis* sample solution and 450 μL of DPPH solution were added to the sample tubes. A total of 50 μL of *A. sinensis* sample solution and 450 μL of absolute ethyl alcohol were added to the sample control tubes. The mixture was mixed and left in the dark at room temperature for 30 min. An appropriate amount of the mixture was used to detect the absorbance value at 517 nm by a microplate reader, which was recorded as A_0_, A_1_, and A_2_ in turn. The DPPH free radical-scavenging rate was calculated according to Equation (1).
DPPH radical scavenging (%) = [A_0_ − (A_1_ − A_2_)]/A_0_ × 100%(1)

Determination of the hydroxyl radical-scavenging rate: the experimental method was operated in strict accordance with the corresponding detection kit instructions. A total of 200 μL of OH assay buffer and 800 μL of distilled water were added to the blank tubes. A total of 150 μL of 1,10-Phenanthroline solution, 200 μL of OH assay buffer, 100 μL of ferrous ion chromogenic solution, and 550 μL of distilled water were added to the undamaged tubes. A total of 150 μL of 1,10-phenanthroline solution, 200 μL of OH assay buffer, 100 μL of ferrous ion chromogenic solution, 450 μL of distilled water, and 100 μL of oxidizing agent were added to the damaged tubes. A total of 200 μL of OH assay buffer, 700 μL of distilled water, and 100 μL of *A. sinensis* sample solution were added to the sample control tubes. A total of 150 μL of 1,10-phenanthroline solution, 200 μL of OH assay buffer, 100 μL of ferrous ion chromogenic solution, 350 μL of distilled water, 100 μL of *A. sinensis* sample solution, 100 μL of oxidizing agent were added to the damaged tubes. The tubes were kept warm in a 37 °C water bath for 60 min. An appropriate amount of the mixture was used to detect the absorbance value at 535 nm by a microplate reader, which was recorded as A_0_, A_1_, A_2_, A_3′_, and A_3_ in turn. The hydroxyl radical scavenging was calculated according to Equation (2).
Hydroxyl radical scavenging (%) = [(A_3_ − A_3′_) − (A_2_ − A_0_)]/(A_1_ − A_2_) × 100%(2)

Determination of the total antioxidant capacity (T-AOC): the experimental method was operated in strict accordance with the corresponding detection kit instructions. In this method, the total antioxidant capacity was expressed as ferrous ion concentration (Fe^2+^ concentration: mM). Therefore, when the total antioxidant capacity of the sample is 1 mM, it means that the substance has the same antioxidant capacity as Fe^2+^ at a concentration of 1 mM. A total of 30 μL of distilled water and 264 μL of FRAP solution were added to the blank tubes. A total of 30 μL of distilled water and 264 μL of FRAP solution were added to the sample tubes. The tubes were kept warm in a 37 °C water bath for 30 min. An appropriate amount of the mixture was used to detect the absorbance value at 593 nm by a microplate reader, which was recorded as A_0_ and A_1_ in turn. The total antioxidant capacity was calculated according to Equations (3) and (4).
Ax = A_1_ − A_0_(3)
y = 1.7473x + 0.1014 (*r* = 0.9999)(4)

When testing the antioxidant activity index for A, S-A, and V-A samples, 6 parallel experiments were performed, and recorded the experimental results. Following the completion of the experiments, the DPPH radical scavenging, hydroxyl radical-scavenging rate, and total antioxidant capacity were calculated using the formula provided in the reagent kit instruction manual.

### 2.6. Statistical Analysis

Multivariate statistical analyses were used to maximize the preservation of original information. Principal component analysis (PCA) is a multi-dimensional data statistical analysis method for unsupervised pattern recognition. At the same time, PCA is also a linear combination of multiple indicators into fewer comprehensive indicators through the idea of dimensionality reduction. PCA can simplify and shrink the data without affecting the analysis results. In this study, R 4.2.2 software was mainly used for statistical analysis to establish numerical models and cluster heat maps for inter-group analysis, which were then used to determine the relevant differences between *A. sinensis* samples.

Subsequently, the Orthogonal Partial Least Squares Method (OPLS-DA) was used to analyze and detect the differences of compounds in *A. sinensis* samples by multivariate statistical analysis method, and a scatter score map and S-plot map were obtained. Finally, the analysis results were substituted into the software for qualitative analysis and identification of the differential biomarkers before and after processing. Hierarchical clustering was used to analyze the differences in metabolite accumulation between samples. The importance of variables in projection (VIP) is a parameter that indicates the importance of variables in the model. It is generally believed that compounds with VIP > 1.0 play an important role in distinguishing groups [17]. Differential metabolites of A, S-A, and V-A were screened by combining the following two criteria. When the VIP of the metabolites in the OPLS-DA model was greater than 1.0, it indicated that the difference between the groups of the corresponding metabolites was significant in the classification and discrimination of each sample in the model. Meanwhile, the *p*-value/false discovery rate (FDR) or fold change (FC) value of multivariate analysis could be used to further screen out the differential metabolites. (For comparison between two groups, metabolites were considered significant if FC ≥ 2 or FC ≤ 0.5; for multiple group comparison, if *p*-value < 0.05, the metabolites were considered as significant).

The tests were repeated in triplicate, and the result of each test is expressed as the average of three replicates. In order to determine the reliability and significance of the data, the measured data were analyzed with one-way analysis of variance (ANOVA) using both SPSS Statistics 24.0 software and GraphPad Prism 9.0 software. Additionally, we utilized gray correlation degree analysis and Pearson’s correlation coefficient analysis to perform a component–effect association analysis between the various components and antioxidant activity in *A. sinensis*.

## 3. Results

### 3.1. QC Sample Analysis

In order to make the results of data acquisition and instrumental analysis more accurate and precise, we evaluated the stability of the evaluation system and analysis method of the experiment. Through the degree of overlap of all QC samples in the PCA score plot, the evaluation signal of correction drift is monitored, and the repeatability and accuracy of the data are monitored to ensure the feasibility of repeated detection of the analyzed samples by experimental technology. The overlapping chart of the total ions chromatogram (TIC) of QC samples (Appendix A) and the QC samples have a very high degree of coincidence, and the results show that the experimental instruments and analytical methods possess excellent stability and repeatability.

### 3.2. Analysis of the Chemical Composition of A. sinensis and Its Concocted Products

In this study, samples were taken from raw *A. sinensis*, fried *A. sinensis*, and vinegar *A. sinensis*, which were analyzed by metabolomics (Appendix A). In the extensive targeted metabolomics analysis, 1046 chemical components were detected in the samples of A, S-A, and V-A in the UPLC-QQQ-MS/MS platform and angelica database, mainly including 11 types of components (Figure 2 and Appendix A): 163 phenolic acids accounted for 15.5%, 153 lipids accounted for 14.9%, 130 amino acids and derivatives accounted for 12.4%, 90 flavonoids accounted for 9.1%, 59 coumarins accounted for 5.6%, 80 organic acids accounted for 7.6%, 75 alkaloids accounted for 7.2%, 60 saccharides accounted for 5.7%, 29 lactones accounted for 2.7%, and so on. Compared with A, the relative content of nucleosides, vitamin, coumarins, phenolic acids, alkaloids, saccharides, and flavonoids were increased, while the relative content of lipids and amino acids were decreased in the processing of S-A. Similarly, compared with A, the relative content of flavonoids, organic acids, vitamins, and lactones increased significantly, while the relative content of aldehyde and ketone, amino acids, and saccharides decreased significantly in V-A. The relative contents of the typical compounds in A, S-A, and V-A were shown in Appendix A.

### 3.3. Multivariate System Analysis

#### 3.3.1. Principal Component Analysis, PCA

PCA was executed on the sample sets A, S-A, and V-A, inclusive of the QC samples (Appendix A). In this analysis, PC1 accounted for a 33.68% variance in the original data across all four groups, while PC2 explained 22.99% of the variance in the *A. sinensis* samples data. These findings reveal a distinct separation trend among A, S-A, V-A, and QC samples. Notably, the QC samples exhibited a consistent aggregation, suggesting that *A. sinensis* and its derivatives were stable during the UPLC-QQQ-MS/MS and metabolomics analysis. The acquired data from its processing methods, chromatography, and mass spectrometry were reliable. The consistency in experimental conditions ensured a high degree of precision.

In a comparative PCA analysis between A and S-A, PC1 explained 52.1% of the variance. Similarly, for A and V-A, PC1 was accounted for 56.89%. PC2 explained 14.42% and 12.42% of the original datasets for A and S-A and A and V-A, respectively. The distinctive patterns were indicated by percentages in *A. sinensis*’s different processing methods, reflecting varied chemical compositions in S-A and V-A post-preparation.

#### 3.3.2. Orthogonal Partial Least Squares Method, OPLS-DA

To delve deeper into the distinctions between sample groups, OPLS-DA was employed. The score plot from this analysis revealed three distinct regions, each indicative of a unique metabolic profile (Appendix A). Group A was distinctly separated from both S-A and V-A. This separation echoes the patterns observed in the PCA, indicating consistent variations in *A. sinensis* treatments over different times.

In the OPLS-DA S-plot (Appendix A), each chemical component’s contribution to *A. sinensis* was evident. Particularly, components with VIP > 1, constituting 433 significantly different components, were highlighted in red. The OPLS-DA model’s verification diagram (Appendix A) indicates high reliability, with R^2^X and Q^2^ values of 0.648 and 0.952 for groups A and S-A and 0.68 and 0.971 for A and V-A. A *p*-value < 0.05 in this model accentuates its strong predictive capability, making it ideal for screening differential components in *A. sinensis* samples.

### 3.4. Analysis of the Differential Composition of A. sinensis and Its Products

Among the 1046 chemical components of *A. sinensis* and its concoction products, a total of 433 differential components were detected, and a total of 402 differential components of S-A and V-A were identified, mainly including 11 types of substances: 16.1% phenolic acids, 14.1% amino acids and their derivatives, 10.1% flavonoids, 9.9% organic acids, 9.9% lipids, 7.4% nucleotides and their derivatives, 7.9% lignin and coumarins, 7.9% alkaloids, 1.7% terpenes, 1.2% quinones, 13.1% other substances. In order to intuitively display the overall metabolic differences between group A and S-A, group A and V-A, and group S-A and V-A, the chemical components of *A. sinensis* and its processed products were screened again by presenting the two indicators of FC and *p*-value in the form of volcano charts (Figure 2 and Appendix A), and the criteria of VIP ≥ 1 and *p* ≤ 0.05 in the OPLS-DA model were combined with FC ≥ 2 or FC ≤ 0.5.

#### 3.4.1. Analysis of the Differential Components of A and S-A

In the cluster heatmap showcasing the differences between raw *A. sinensis* and fried *A. sinensis*, we identified a total of 102 differential components in groups A and S-A. Specifically, 11 components exhibited downregulation, while 91 were upregulated (Figure 2C). These differential components were chiefly categorized into 10 classes (Appendix A), comprising amino acids and their derivatives (16 upregulated), phenolic acid components (15 upregulated, 1 downregulated), alkaloids (11 upregulated, 1 downregulated), coumarins (10 upregulated), terpene components (9 upregulated), organic acid components (7 upregulated), flavonoids (5 upregulated, 2 downregulated), lignans (1 upregulated), lipids (2 upregulated, 3 downregulated), and nucleotides with their derivatives (4 upregulated, 2 downregulated).

Subsequently, based on the FC magnitude, the top 10 upregulated and 10 downregulated differential compounds were selected for plot analysis (Appendix A). Among these 20 differential components in groups A and S-A (Table 1), amino acids and their derivatives constituted 70%. During high-temperature stir-frying, there was an increase in the relative content of compounds like 3-hydroxyquinidine and dihydrocaffeic acid in alkaloids and 6’-*O*-cinnamyl-8-epi-genotyl in the phenolic acid component. In contrast, the relative content of several compounds, including phloroglucinol, 1,3,5-benzenetriol (phenolic acid), and others listed, were decreased. These observations suggest that the high-temperature stir-frying process aids in the synthesis and degradation of these components.

The radar plots were created using the top 10 differential components and analyzed based on the absolute size of log_2_FC in groups A and S-A, which are presented in Appendix A. This analysis highlighted components such as 6’-*O*-cinnamoyl-8-epi-arygioside, 3-hydroxyquinidine, dihydrocaffeoylputrescine, and others. Notably, amino acids and their derivatives represented 70% of these compounds. The relative contents of these 10 distinct components demonstrated an upward trend during stir-frying, indicating the process’s role in enhancing the formation of these differential components.

#### 3.4.2. Analysis of the Differential Components of A and V-A

A comprehensive analysis revealed 167 differential components between group A and V-A (Figure 2D). Specifically, within V-A, 43 components were downregulated, while 124 exhibited upregulation (Appendix A). These components predominantly fell into categories such as amino acids and their derivatives (27 upregulated, 5 downregulated), alkaloids, flavonoids, and phenolic acids, among others. Among the top 10 components with a notably lower relative were contented in groups A and V-A, two compounds each were identified from amino acids and their derivatives (namely L-methionine methyl ester and serine–glycine), phenolic acids, and alkaloids. Other identified differentiating components spanned categories, including flavonoids, lipids, vitamins, and other components. These findings suggest that an acidic environment combined with high-temperature frying conditions favors the synthesis and degradation of these components (Appendix A). In evaluating components based on the absolute magnitude of log_2_FC (Table 2), the results showed a dominance of amino acids and their derivatives, with a few alkaloids, phenolic acids, and one other component (Appendix A). Among these, only ethyl 1,4-di-*O*-caffeoylquinate displayed a significant reduction in relative content. Conversely, components like methyl 2,4-dihydroxyphenylacetate, cyclo(proline–proline), and others showed an upward regulatory trend. This suggests that an acidic environment coupled with high-temperature processing particularly enhances the formation of amino acids and their derivatives.

Furthermore, a significant increase in the relative content of certain components was observed during the stir-frying and vinegar treatments. These components, such as 6’-*O*-cinnamoyl-8-epi-arygioside, 3-hydroxyquinidine, and others, could play a pivotal role in enhancing the antioxidant and lipid-lowering properties of fried *A. sinensis* and vinegar *A. sinensis*.

#### 3.4.3. Analysis of the Common Differential Components

In the comparison between group A with S-A and group A with V-A, 83 common differential components were identified (Figure 3A, Appendix A). These shared components hint at the underlying chemical similarities between S-A and V-A. The breakdown included 16 phenolic acids, 15 amino acids, and their derivatives, and 12 alkaloids, among others. Notably, high-temperature processing of *A. sinensis* introduced two new chemical entities: *N*-pentadecanoyl-L-homoserine lactone (an amino acid derivative) and 6’-*O*-Cinnamoyl-8-epi-arygioside (a phenolic acid). These unique compounds are exclusive to both S-A and V-A. Of these differentiated components, phenolic acids dominate, comprising 19.28% (refer to Appendix A, Table 1 and Table 2). Only phloroglucinol and 1,3,5-benzenetriol displayed marked variations in their contents during processing (*p* < 0.05), while other phenolic acids like benzamide increased. Amino acids and their derivatives represent the second most abundant class at 18.07%. Most of these showed increased content, especially the dipeptides. However, *N*2-(1-Carboxyethyl)-L-arginine increased in S-A but decreased in V-A.

In the nucleotide and derivative category, only the contents of *N*-(1-deoxy-1-fructosyl) tryptophan and isopentenyladenine-7-*N*-glucoside were declined. The flavonoids luteolin-7-*O*-glucuronide and kaempferol 7-*O*-glucoside significantly decreased (*p* < 0.01) and participated in the Maillard reaction with amino compounds [18]. The coumarin and organic acid components trended upward during processing, suggesting that high temperatures favor their synthesis. V-A had the highest content of coumarins, followed by S-A and A. Among alkaloids, only 1-*O*-caffeoyllysine’s content was dropped during high-temperature treatment. In other categories, compounds like ribulose-5-phosphate and ethyl 1,4-di-*O*-caffeoylquinate showed a marked decrease (*p* < 0.01). Aldehydes, specifically 5-HMF and 5-methoxyfurfural, significantly increased. The quality and functional activity of *A. sinensis* and its thermally processed variants were influenced by these 83 differential components.

### 3.5. In Vitro Antioxidant Activity Studies

The antioxidant activity of *A. sinensis* is commonly evaluated using the DPPH free radical-scavenging rate, hydroxyl free radical-scavenging rate, and T-AOC. The effects of processing on the antioxidant activity were evident in the changes in these metrics (Table 3).

For the DPPH free radical-scavenging rate, the rates for A, S-A, and V-A were recorded at 48.77%, 49.09%, and 60.80%, respectively. A significant difference was observed among these *A. sinensis* samples (*p* < 0.01). Notably, the DPPH scavenging rate was considerably enhanced by vinegar processing, whereas stir-frying (S-A) showed no significant change.

The hydroxyl free radical-scavenging rates for A, S-A, and V-A were 71.32%, 75.84%, and 67.68%, respectively. The order of effectiveness was S-A > A > V-A, with significant disparities observed (*p* < 0.01). The hydroxyl radical-scavenging rate was increased in stir-frying processing, while a slight reduction resulted in vinegar processing (V-A).

The T-AOC was determined by the FRAP method to vary among the processed products. The T-AOC values followed the order S-A > V-A > A, with each demonstrating significant differences (*p* < 0.01). Post-processing, the T-AOC was increased in all samples. S-A had the highest T-AOC, while V-A trailed closely behind.

### 3.6. Composition–Effectiveness Relationship Analysis

The gray correlation degree analysis is a multivariate statistical method that is straightforward to comprehend and apply, especially when handling data with multiple factors that do not exhibit typical distribution characteristics [17,19]. This method, when setting the distinguishing coefficient at 0.5, determines the correlation between the relative peak areas of 83 distinct components on the chromatogram and the antioxidant activity in *A. sinensis*. A stronger relationship between the two indexes was implied by a higher correlation coefficient. The antioxidant activity of *A. sinensis* was assessed using this gray correlation method (refer to Appendix A). Components with a gray relational coefficient exceeding 0.7 were deemed significant contributors to antioxidant activity. Consequently, 62 compounds displayed a strong correlation with DPPH free radical scavenging, while 46 and 61 compounds exhibited significant activities related to hydroxyl free radical scavenging and total antioxidant capacity, respectively. Overall, 44 compounds demonstrated a high correlation degree (>0.7) concerning their antioxidant capacity post-preparation. Seven of these compounds stood out: salicylic acid, 4-hydroxybenzaldehyde, osthenol, 4-acetamidobutyric acid, imocitrin-7-*O*-glucoside, syringetin-3-*O*-glucoside, and angeloyl senkyunolide F. Each had scores exceeding 0.79 and displayed robust correlation with all three antioxidant activity indicators, suggesting their significant role in neutralizing free radicals and enhancing antioxidant activity.

To deepen our understanding of the relationship between differential compounds and antioxidant activity, the identified 44 compounds were executed by Pearson’s correlation analysis. Here, red and blue were used to symbolize positive and negative correlations between compounds and antioxidant activity, respectively. As illustrated in Figure 3B, most of the 44 compounds were positively correlated with DPPH free radical scavenging. However, three compounds, *N*-(1-deoxy-1-fructosyl)tryptophan, luteolin-7-*O*-glucuronide, and ribulose-5-phosphate, exhibited a negative correlation. Among these, the raw material (A) presented the strongest positive association with DPPH free radical scavenging. For hydroxyl free radical scavenging, the correlation coefficients for most compounds were near zero, suggesting a minimal contribution. Osthenol, 2-methyl-5,7,8-trimethoxyisoflavone, pyrrole-2-carboxylic acid, 5-aminovaleric acid, and *β*-ureidoisobutyric acid displayed weak positive correlations. In terms of total antioxidant capacity, 41 compounds showed a positive correlation, while 3-*N*-(1-deoxy-1-fructosyl)tryptophan, luteolin-7-*O*-glucuronide, and ribulose-5-phosphate showed a negative correlation. Interestingly, *β*-ureidoisobutyric acid exhibited the highest positive correlation with total antioxidant capacity.

## 4. Discussion

Utilizing the UPLC-QQQ-MS/MS coupled with advanced targeted metabolomics technology, we dissected the chemical composition of *A. sinensis* and its subsequent processed products. A total of 1046 chemical components were detected by this comprehensive analysis, with 402 distinguishing components categorized under 11 notable classes: quinones, terpenoids, nucleotides and their derivatives, alkaloids, organic acids, lignans and coumarins, flavonoids, amino acids and their derivatives, lipids, phenolic acids, and others. Significantly, our findings highlight the pronounced chemical alterations in *A. sinensis* post-processing. The relative contents of several compounds were reduced after processing, namely quinones, nucleotides, organic acids, flavonoids, and lipids. Conversely, the relative content of vitamins, catechins, and terpenes was significantly increased.

High-temperature processing especially seems to impact amino acids and their derivatives alongside carbohydrates. As processing progresses, many amino acids evolve into small-molecule dipeptides, including the likes of tryptophane–alanine, isoleucine–leucine, and glutamic acid–phenylalanine. The significance of proteins, which consist of amino acid residues, in food cannot be overstated [20,21]. They underpin numerous biophysical functions and dictate the nutritional, sensory, and longevity aspects of food items. Amino acids, essential for human nutrition, also laid the foundation for peptides and proteins [22]. *A. sinensis*, inherently protein-rich, undergoes structural and functional protein modifications under extreme conditions like temperature fluctuations, changes in pH, and varying moisture content [23]. Elevated temperatures and moisture loss, for instance, can be dismantled by weak proteins and structured into tinier compounds. In acidic conditions, proteins and polypeptides face disruptions, simplifying their decomposition into smaller amino acid molecules and consequently amplifying their relative abundance. However, certain amino acids such as histidine, leucine, valine, glutamic acid, and tryptophane experienced a notable drop post-processing.

Processed carbohydrates primarily transitioned into reducing sugar and alduronic acids, including D-galacturonic and D-glucoronic acids. But, levels of D-xylonic, D-galacturonic acid, D-glucoronic acid, and Glucose-1-phosphate dwindled in the processing. Similarly, several carbonyl compounds, ferulic acid, senkyunolide H, adenine, coumarin, kaempferol 7-*O*-glucoside, and nicotinic acid, among others, saw diminished relative content (Figure 4). In contrast, the content of 5-hydroxymethylfurfural (5-HMF) registered a prominent rise in the chemical composition of *A. sinensis*.

Our meticulous analysis spotlighted 102 differential components between groups A and S-A, and 167 between A and V-A. These distinctions potentially drive the varying flavor profiles and pharmacological impacts across the three groups. Interestingly, saccharides and amino acids, producing pyrazines, ketones, and 5-HMF, were reduced during the Maillard reaction, which was responsible for the distinctive roasted aroma of *A. sinensis* during processing (Figure 5). Organic acids and phenolic acid components not only dictate the sensory aspects of *A. sinensis* but also proffer health benefits, including radical clearance and blood rheology improvement. Likewise, flavonoids, alkaloids, lignans, and coumarins showcase antioxidant prowess and have implications in managing cardiovascular and cerebrovascular ailments. Numerous other components, primarily polysaccharides, exhibit myriad biological activities, from antioxidant to lipid-lowering effects [24]. Notably, the content of 5-HMF escalated dramatically during the processing, registering a 15-fold and 20-fold increase in S-A and V-A, respectively. 5-HMF’s vast antioxidative capabilities are well documented [25]. It effectively neutralizes several free radicals and shields the body against oxidative damage stemming from high glucose, alcohol, and ischemia. Furthermore, 5-HMF’s positive influence on blood health accentuates the circulatory and nourishing properties of *A. sinensis*.

Chinese medicinal materials, which possess both therapeutic and culinary properties, were processed and underwent an intricate Maillard reaction that is markedly more complex than what is typically observed during regular food processing [18]. This reaction is evident as these medicinal pieces take on a darker hue, often blackening, while exuding an aromatic fragrance, hallmark traits of the Maillard reaction. Interestingly, this reaction also leads to the genesis of novel compounds with pharmacological efficacy [26]. For instance, under the influence of high-temperature processing, the deepened color and the smell of charmingly burned were owned by *A. sinensis*. This is indicative of certain carbonyl and amino compounds, spanning amino acids, peptides, and proteins, engaging in the Maillard reaction when exposed to high temperatures (Figure 5) [26]. This dynamic chemical equilibrium resulted in the significant generation of 5-hydroxymethylfurfural [27], which reshaped the nutrient framework and flavor spectrum of *A. sinensis*. 

Within this processing landscape, acidic conditions catalyze the Amadori rearrangement of sugar-laden substances present in *A. sinensis*. When these rearranged entities are subjected to heightened temperatures, they participate in the Maillard reaction alongside carbonyl and amino compounds. Chemical components were interplay births a plethora of intermediary products, such as restoring ketone compounds, aldehyde compounds, and aromatic heterocyclic entities. These compounds were pivotal in crafting the aromatic signature of the processed material [28]. Progressing through the Maillard reaction’s stages, melanoidin emerges as the culmination product, renowned for its potent antioxidant prowess. Indeed, the by-products in the Maillard reaction were instrumental in dictating the visual and olfactory transformations observed in processing *A. sinensis* [29,30]. Within this realm, 5-HMF, which fell under the aldehydes, is intrinsically tied to both caramelization and Maillard processes [30]. 5-HMF’s origin can be traced to the dehydration of saccharides in acidic milieus, facilitated by caramelization and Maillard mechanisms [31,32] (Figure 5). A sizable portion of 5-HMF can metamorphose further, birthing downstream compounds. An exemplar is succinic acid. However, succinic acid was structurally temperamental [33], readily forfeiting water during high-temperature exposure, leading to the abundant formation of succinic anhydride (Figure 4). Within the processing framework of *A. sinensis*, a pronounced upsurge in succinic anhydride concentration is evident. This indicated a fervent Maillard reaction, culminating in the formation of 5-HMF, which then undergoes further transformations. This results in the emergence of succinate, which was exposed to elevated temperatures and swiftly degrades to succinic anhydride, thereby boosting its concentration.

In encapsulation, high-temperature processing either synthesizes or disassembles specific chemical constituents within *A. sinensis*. The Maillard reaction stands out as a cornerstone in the high-temperature processing of *A. sinensis*, steering the alterations observed in its chemical and aromatic profile post-processing. Broadly, the shifts in relative content and qualitative changes in the core chemical constituents of A, S-A, and V-A were delineated by this exposition.

## 5. Conclusions

In the present study, we employed a combination of UPLC-QQQ-MS/MS and targeted metabolomics to dissect the chemical composition shifts in *A. sinensis* and its processing products. A comprehensive profile of 1046 chemical entities, spanning a spectrum of terpenoids, quinones, nucleotides, flavonoids, lignans, coumarins, and many others, was established. It became evident that processing indeed brings about significant changes in the metabolic profile of *A. sinensis*. Particularly, 102 differential components were detected in the S-A, while 167 differential components in the V-A and 83 differential components were detected in stir-frying and vinegar-frying. Such variations could be pivotal in the discernible differences in flavor and antioxidant attributes between A, S-A, and V-A. The chemical constituents in *A. sinensis* undergo diverse changes upon stir-frying and vinegar processing, as revealed by the analysis. Specifically, in the V-A sample, predominant components like 1-*O*-caffeoyllysine and ethyl 1,4-di-*O*-caffeoylquinate witnessed a decline, while amino acids, phenolic acids, alkaloids, and coumarins experienced significant increments, especially after stir-frying in the presence of garlic and vinegar. This disparity between S-A and V-A could be credited to the introduction of rice vinegar before the high-temperature treatment. Post-processing, a bolstered antioxidant potential was showcased in *A. sinensis*, reflected in the enhancement of various antioxidant activity markers. Compounds like 4-hydroxybenzaldehyde and osthenol, among others, had a pronounced influence on the antioxidant prowess of *A. sinensis* throughout its processing.

By tracing the trajectory of chemical changes in *A. sinensis* processing, we unraveled the underlying pathways of the Maillard reaction during high-temperature treatments. Establishing correlations between the chemical components and antioxidant markers, a strong positive association was observed between the most varied chemical components and antioxidant activity during processing. A scientific groundwork for comprehending the transformative dynamics of *A. sinensis* under diverse processing regimes was cemented by this research. The way for a richer understanding of the *A. sinensis* processing mechanism was paved, while a theoretical scaffold to assess the interplay between differentiating components and their antioxidant activity during processing was concurrently offered. As a forward-looking endeavor, the links between these differentiating components and the multitude of pharmacological activities were exhibited by processed *A. sinensis*, which was probed in future investigations.

## Figures and Tables

**Figure 1 antioxidants-12-02053-f001:**
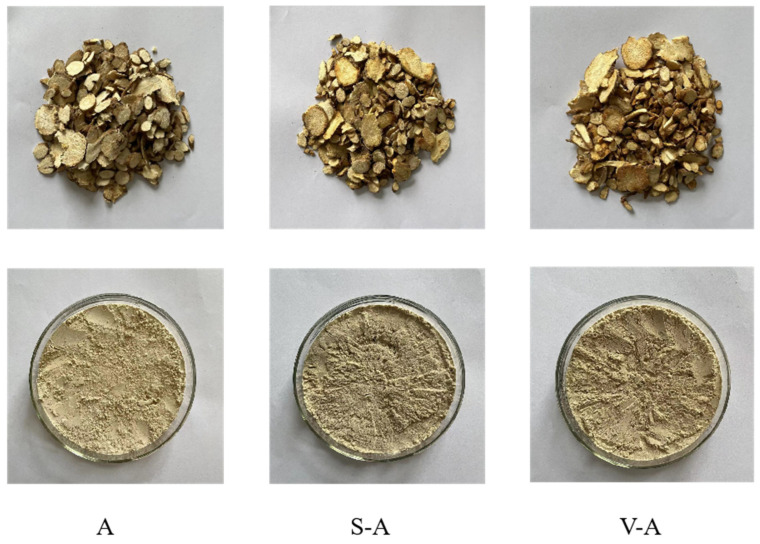
Picture of A, S-A, and V-A.

**Figure 2 antioxidants-12-02053-f002:**
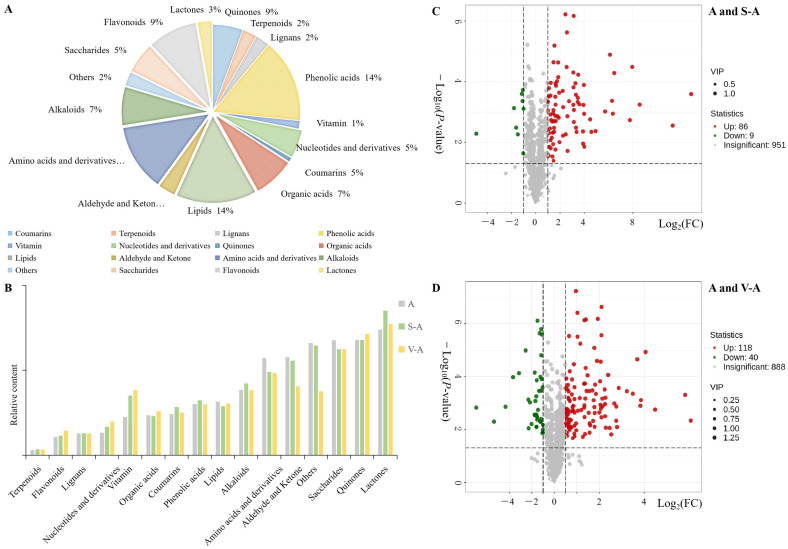
Pie chart of the proportion of each substance in the chemical components of *A. sinensis* (**A**), its processed products and bar chart of the relative content proportion (**B**), and Volcano plots of differential components between A and S-A (**C**) and between A and V-A (**D**). Note: Red indicates upregulation, and green indicates downregulation.

**Figure 3 antioxidants-12-02053-f003:**
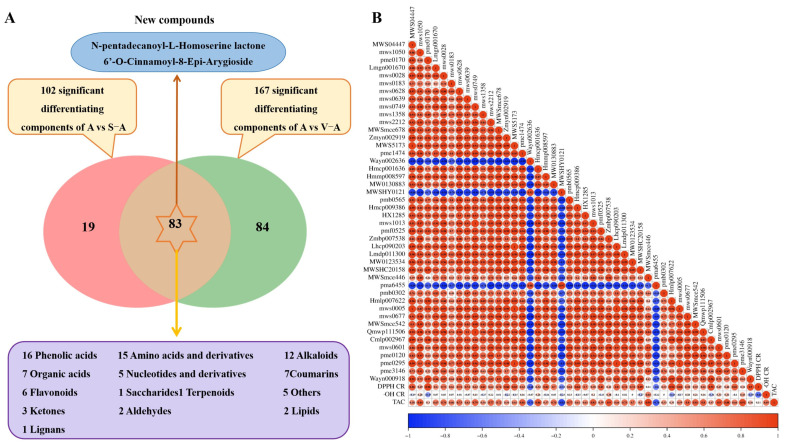
Venn diagram of different components of A and S-A, A, and V-A (**A**) and Pearson’s correlation coefficient analysis between 44 differential compounds and antioxidant activity (**B**).

**Figure 4 antioxidants-12-02053-f004:**
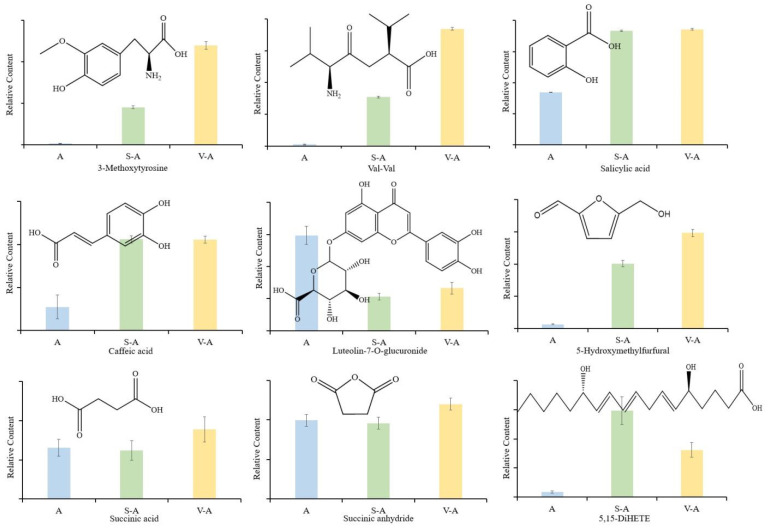
Changes of 9 differential compounds in *A. sinensis* during processing.

**Figure 5 antioxidants-12-02053-f005:**
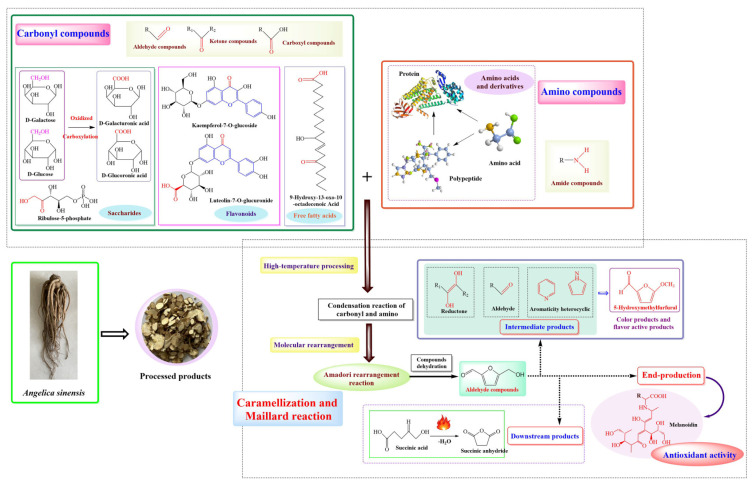
Change patterns in the main types of components in *A. sinensis* during processing.

**Table 1 antioxidants-12-02053-t001:** Top 10 compounds with absolute values of FC in the upregulated and downregulated differential components of A and S-A.

Group	Index	Formula	Compounds	Class I	A	S-A	VIP	*p-*Value	FC	UP/Down
A and S-A	MW0105650	C_10_H_20_N_6_O_4_	Argnine–Asparagine	Amino acids and derivatives	2.13 × 10^4^	1.75 × 10^6^	1.38	1.13 × 10^−3^	1.22 × 10^−2^	up
	MW0169576	C_19_H_35_NO_3_	*N*-pentadecanoyl-L-Homoserine lactone		9.00	2.26 × 10^4^	1.39	2.78 × 10^−3^	3.99 × 10^−4^	up
	MW0159268	C_11_H_18_N_4_O_3_	Valine–Histidine		3.26 × 10^4^	7.07 × 10^6^	1.39	1.84 × 10^−3^	4.61 × 10^−3^	up
	MW0151674	C_12_H_24_N_2_O_3_	Isoleucine–Leucine		4.03 × 10^3^	3.58 × 10^5^	1.39	5.19 × 10^−5^	1.12 × 10^−2^	up
	MW0105085	C_10_H_13_NO_4_	3-Methoxytyrosine		5.77 × 10^3^	4.51 × 10^5^	1.35	4.27 × 10^−4^	1.28 × 10^−2^	up
	MW0110378	C_10_H_20_N_2_O_3_	Valine–Valine		1.78 × 10^3^	1.24 × 10^5^	1.36	1.29 × 10^−5^	1.44 × 10^−2^	up
	Lmrj002244	C_10_H_14_N_2_O_2_	Cyclo(Proline–Proline)		1.32 × 10^6^	6.95 × 10^5^	1.38	9.59 × 10^−4^	1.90 × 10^−2^	up
	Smnp005894	C_25_H_28_O_12_	6’-*O*-Cinnamoyl-8-Epi-Arygioside	Phenolic acids	9.00	6.49 × 10^6^	1.39	2.55 × 10^−4^	1.39 × 10^−4^	up
	NK10264324	C_6_H_6_O_3_	Phloroglucinol; 1,3,5-Benzenetriol		1.32 × 10^6^	2.37 × 10^5^	1.31	1.05 × 10^−1^	5.54	down
	MW0000084	C_20_H_24_N_2_O_3_	3-Hydroxyquinidine	Alkaloids	4.58 × 10^3^	1.74 × 10^6^	1.38	5.71 × 10^−4^	2.64 × 10^−3^	up
	Lmmp001410	C_13_H_20_N_2_O_3_	Dihydrocaffeoylputrescine		7.41 × 10^3^	1.84 × 10^6^	1.39	3.26 × 10^−5^	4.03 × 10^−3^	up
	Wafp002310	C_15_H_20_N_2_O_5_	1-*O*-Caffeoyllysine		1.83 × 10^5^	5.97 × 10^5^	1.38	3.26 × 10^−3^	3.07	down
	Wayn002636	C_17_H_22_N_2_O_7_	*N*-(1-Deoxy-1-fructosyl)Tryptophan	Nucleotides and derivatives	2.57 × 10^6^	1.21 × 10^6^	1.37	4.33 × 10^−4^	2.13	down
	pmb0964	C_16_H_23_N_5_O_5_	Isopentenyladenine-7-*N*-glucoside		7.06 × 10^5^	2.17 × 10^5^	1.23	6.47 × 10^−2^	3.26	down
	MWSHY0121	C_21_H_18_O_12_	Luteolin-7-*O*-glucuronide	Flavonoids	7.43 × 10^5^	2.67 × 10^5^	1.36	5.43 × 10^−3^	2.78	down
	MW0138670	C_21_H_20_O_11_	Kaempferol 7-*O*-glucoside		1.08 × 10^5^	3.60 × 10^4^	1.38	5.14 × 10^−3^	3.00 × 10^1^	down
	pma6455	C_5_H_11_O_8_P	Ribulose-5-phosphate	Saccharides	2.53 × 10^5^	1.24 × 10^5^	1.34	2.29 × 10^−4^	2.04	down
	Lsmp010410	C_27_H_28_O_12_	Ethyl 1,4-di-*O*-caffeoylquinate	Others	3.27 × 10^5^	9.43 × 10^5^	1.38	7.43 × 10^−4^	3.47	down
	Lmhp012042	C_21_H_38_O_4_	2-Linoleoylglycerol	Lipids	2.99 × 10^5^	1.43 × 10^5^	1.37	1.88 × 10^−4^	2.09	down
	Rfmb087	C_18_H_32_O_4_	9-Hydroxy-13-oxo-10-octadecenoic acid		4.94 × 10^5^	2.23 × 10^5^	1.38	2.51 × 10^−4^	2.22	down

**Table 2 antioxidants-12-02053-t002:** Top 10 compounds with absolute values of FC in the upregulated and downregulated differential components of A and V-A.

Group	Index	Formula	Compounds	Class I	A	V-A	VIP	*p*-Value	FC	Type
A and V-A	Lmrj002244	C_10_H_14_N_2_O_2_	Cyclo(Proline–Proline)	Amino acids and derivatives	2.23 × 10^5^	4.94 × 10^5^	1.32	3.56 × 10^−4^	8.73 × 10^1^	up
	MW0110378	C_10_H_20_N_2_O_3_	Valine–Valine		2.23 × 10^5^	4.94 × 10^5^	1.31	2.21 × 10^−5^	1.66 × 10^2^	up
	MW0159268	C_11_H_18_N_4_O_3_	Valine–Histidine		2.23 × 10^5^	4.94 × 10^5^	1.33	1.27 × 10^−3^	2.01 × 10^2^	up
	MW0105085	C_10_H_13_NO_4_	3-Methoxytyrosine		2.23 × 10^5^	4.94 × 10^5^	1.30	7.71 × 10^−4^	2.08 × 10^2^	up
	pme1419	C_6_H_13_NO_2_S	L-Methionine methyl ester		2.2 × 10^5^	4.94 × 10^5^	1.33	1.04 × 10^−4^	7.84 × 10^−2^	down
	MW0109707	C_5_H_10_N_2_O_4_	Serine–Glycine		2.23 × 10^5^	4.94 × 10^5^	1.32	7.38 × 10^−5^	1.13 × 10^−1^	down
	MW0169576	C_19_H_35_NO_3_	*N*-pentadecanoyl-L-Homoserine lactone		2.23 × 10^5^	4.94 × 10^5^	1.33	4.87 × 10^−4^	3.17 × 10^3^	up
	Hmtn001288	C_9_H_10_O_4_	Methyl 2,4-Dihydroxyphenylacetate	Phenolic acids	2.23 × 10^5^	4.94 × 10^5^	1.32	4.44 × 10^−4^	1.26 × 10^2^	up
	Smnp005894	C_25_H_28_O_12_	6’-*O*-Cinnamoyl-8-Epi-Arygioside		2.23 × 10^5^	4.94 × 10^5^	1.33	4.60 × 10^−3^	4.49 × 10^3^	up
	Zmhn002508	C_21_H_28_O_12_	4-*p*-Cumaroyl-rhamnosyl-(1 → 6)-D-glucose		2.23 × 10^5^	4.94 × 10^5^	1.32	1.02 × 10^−5^	1.71 × 10^−1^	down
	NK10264324	C_6_H_6_O_3_	Phloroglucinol; 1,3,5-Benzenetriol		2.23 × 10^5^	4.94 × 10^5^	1.23	1.21 × 10^−1^	2.44 × 10^−1^	down
	Smcp000882	C_15_H_21_NO_7_	*N*-benzoyl-2-aminoethyl-*β*-D-glucopyranoside	Alkaloids	2.23 × 10^5^	4.94 × 10^5^	1.32	2.69 × 10^−4^	5.30 × 10^1^	up
	Lmmp001410	C_13_H_20_N_2_O_3_	Dihydrocaffeoylputrescine		2.23 × 10^5^	4.94 × 10^5^	1.32	1.17 × 10^−5^	2.77 × 10^2^	up
	MW0000084	C_20_H_24_N_2_O_3_	3-Hydroxyquinidine		2.23 × 10^5^	4.94 × 10^5^	1.32	1.77 × 10^−3^	4.98 × 10^2^	up
	Wafp002310	C_15_H_20_N_2_O_5_	1-*O*-Caffeoyllysine		2.23 × 10^5^	4.94 × 10^5^	1.32	1.37 × 10^−3^	4.93 × 10^−2^	down
	Lshp011101	C_20_H_23_NO_12_	Narciclasine 4-Glucopyranoside		2.23 × 10^5^	4.94 × 10^5^	1.32	9.43 × 10^−4^	2.23 × 10^−1^	down
	Lsmp010410	C_27_H_28_O_12_	Ethyl 1,4-di-*O*-caffeoylquinate	Others	2.23 × 10^5^	4.94 × 10^5^	1.32	1.49 × 10^−3^	8.11 × 10^−3^	down
	MA10039492	C_6_H_6_O_6_	Dehydroascorbic acid	Vitamin	2.23 × 10^5^	4.94 × 10^5^	1.32	8.96 × 10^−3^	2.05 × 10^−1^	down
	Rfmb087	C_18_H_32_O_4_	9-Hydroxy-13-oxo-10-octadecenoic acid	Lipids	2.23 × 10^5^	4.94 × 10^5^	1.31	7.42 × 10^−4^	2.01 × 10^−1^	down

**Table 3 antioxidants-12-02053-t003:** DPPH free radical scavenging, hydroxyl free radical scavenging, and T-AOC of *A. sinensis* and its processed products.

Detection Metrics	A	S-A	V-A
DPPH free radical scavenging (%)	46.30 ± 0.75	49.63 ± 1.07 **	60.65 ± 1.06 **
Hydroxyl free radical scavenging (%)	70.39 ± 1.04	76.52 ± 0.97 **	66.60 ± 0.97 **
T-AOC (nM)	0.33 ± 0.01	0.41 ± 0.01 **	0.36 ± 0.01 **

Note: *: indicates the significance level of the ANOVA. **: *p* < 0.01 for processed products compared with A.

## Data Availability

Data will be made available on request.

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
