# Peer review of "Targeted Metabolomics Study on the Effect of Vinegar Processing on the Chemical Changes and Antioxidant Activity of Angelica sinensis"

_antioxidants, 2023, doi:10.3390/antiox12122053_

Round 1

Reviewer 1 Report

Comments and Suggestions for Authors

The manuscript includes an interesting study. It is well presented and justified. Additionally, it includes a combination of advanced analytical tools. I think it can be accepted for publication provided some minor aspects are clarified.

Title

It is too long. Some shortening could be: Targeted metabolomics study on the effect of vinegar processing on the chemical changes and antioxidant activity of Angelica sinensis.

Abstract

Lines 18-21: Maybe some shortening could be done on these general aspects.

No indication is provided regarding the procedure. Should it be modified in order to obtain more valuable composition and higher antioxidant behaviour ?

Material and methods

Why 70% methanol was employed ? The authors ought to include some justification.

Line 93: This section is written as if a culinary recipe was concerned. The authors ought to provide information according to the scientific nature of the journal.

Line 118: Express rpm in xg units.

Comments on the Quality of English Language

Some minor performances could be done.

Reviewer 2 Report

Comments and Suggestions for Authors

The authors Linlin Chen et al. describe a “Study on the changes of the chemical composition and its antioxidant activity of Angelica sinensis with vinegar processing based on widely targeted metabolomics”.

The study is very complex and difficult to read because it is not clearly detailed and the experimental part is not adequately described.

Many points need to be corrected

First, correct all Angelica sinensis in italic Angelica sinensis

Page 1, line 38: “Ref [1] this reference does not seem adequate

Page 4 Table 1. Table 1 is not clear and abbreviations should be supported by an explanation. * unclear.

Page 6 “2.5. In vitro Antioxidant Activity Assay”: This paragraph is meaningless. It gives no indication of the methodology used. If the entire study is based on increasing the antioxidant activity of S-A and V-A, this paragraph should be expanded.

Page 6, lines 199 and 200: “Using the method of multivariate statistical analysis, we analyzed the data collected in section 2.6.” should be removed.

Page 7, line 245: “The total ion chromatograpy (TIC) of QC samples (Fig. S1)” is a mistake?

Page 7, line 264-265: “The relative content of lignans and coumarins, organic acids, phenolic acids, alkaloids… in V-A increased significantly” is not evident from the figure 2B.

Paragraphs 3.3., 3.3.1, 3.3.2 are repeated in the text creating confusion. please correct.

Without going into the merits of multivariate data analysis and targeted metabolomics, the conclusions are based on an investigation of the antioxidant activity of the various derivatives analysed, which is not very pronounced, in my opinion.

Comments on the Quality of English Language

The work is very articulate but needs to be revised in some points

Round 2

Reviewer 2 Report

Comments and Suggestions for Authors

New version of the  manuscript "Targeted metabolomics study on the effect of vinegar  processing on the chemical changes and antioxidant activity of Angelica sinensis" is improved.

Some corrections:

Line 263: "orthogonal" please correct in Orthogonal

Line 333: "Figure (S1)" please correct in Figure (S2)

Comments on the Quality of English Language

The quality of English is improved

Author Response

  1. Line 263: "orthogonal" please correct in Orthogonal

Response: Sorry for our carelessness. We rechecked the manuscript, and "orthogonal" have been changed to "Orthogonal" in the manuscript. Please see line 269.

  1. Line 333: "Figure (S1)" please correct in Figure (S2)

Response: Sorry for our carelessness. We rechecked and corrected in Figure (S2). Please see line 340. In addition, we have thoroughly checked the language, formatting, and other overlooked areas, and the modifications have been highlighted in red.